# Cryopreserved Tissue Biospecimens Offer Superior Quality for Whole-Genome Sequencing of Various Cancers Compared to Paired Formalin-Fixed Paraffin-Embedded Tissues

**DOI:** 10.3390/ijms262211038

**Published:** 2025-11-14

**Authors:** Ken Dixon, Jeong-Hoon Lee, Ryan Miller, David Booker, DeLaney Anderson, Jeffrey Okojie, Matthew Kirkham, Eun Kyoung Lee, Chunyang Bao, Islam Oguz Tuncay, Jung-Ah Kim, Sangmoon Lee, Jared Barrott

**Affiliations:** 1Specicare Inc., Gainesville, GA 30501, USA; ken.dixon@specicare.com (K.D.);; 2Inocras Inc., San Diego, CA 92121, USAoguztuncay@inocras.com (I.O.T.);; 3Department of Internal Medicine and Liver Research Institute, Seoul National University College of Medicine, Seoul 03080, Republic of Korea; 4Department of Cell Biology and Physiology, Brigham Young University, Provo, UT 84602, USAjokojie@student.byu.edu (J.O.); 5Department of Ophthalmology, Seoul National University College of Medicine, Seoul 03080, Republic of Korea; 6Simmons Center for Cancer Research, Brigham Young University, Provo, UT 84602, USA

**Keywords:** whole-genome sequencing, cancer, cryopreservation, FFPE, preanalytical variables

## Abstract

Whole-genome sequencing (WGS) is integral to precision oncology, yet most cancer biospecimens used for WGS are formalin-fixed paraffin-embedded (FFPE) due to their widespread availability in clinical practice. However, FFPE processing can degrade DNA quality. This study compares WGS outcomes from matched cryopreserved (CP) and FFPE tumor samples, hypothesizing that CP tissues yield superior sequencing quality and variant detection. Fifty matched pairs of CP and FFPE tumor samples spanning multiple cancer types were obtained from a biobank. DNA was extracted, and WGS was performed. We assessed sequencing quality metrics and variant analysis between the two preservation methods. Presequencing metrics favored CP tissue, with a significantly higher gDNA concentration, DIN, and DNA fragment size. The WGS results showed that the CP samples had a higher mean read depth and larger insert size. Although the mapping percentages were similar, FFPE exhibited higher tumor mutation burden (13.7 vs. 6.4 mutations/Mb) and lower concordance with CP in variant calls (43.5% overlap). CP samples detected more structural variants and enabled the improved identification of oncogenic driver mutations. Cryopreserved tissues consistently outperform FFPE in terms of DNA quality and WGS metrics, enabling the more accurate detection of clinically relevant mutations. These findings support prioritizing CP sample preservation for genomic profiling in cancer care.

## 1. Introduction

Cancer presents a substantial global clinical burden, together with the high expenses associated with oncological treatments. According to the estimation by GLOBPCAN, there were approximately 20 million new cancer cases and 9.7 million cancer-related deaths worldwide in 2022 [1]. The estimates indicate that roughly 20% of both men and women will develop cancer at some point in their lives, whereas approximately 11% of men and 8% of women will die from it. In the United States, approximately 2 million new cases of cancer are projected to be diagnosed, resulting in approximately 600,000 deaths attributable to cancer in 2024 [2]. This increasing incidence of cancer and the resulting socioeconomical strain highlight the importance of precision medicine.

Precision medicine, primarily enabled by next-generation sequencing (NGS), including whole-genome sequencing (WGS), can offer notable advantages in the field of cancer therapy [3,4]. Precision medicine decreases the use of ineffective therapies and the accompanying healthcare expenses by tailoring personalized treatments based on individual genomic profiles. Targeted treatments selected by precision medicine result in improved patient outcomes, such as increased progression-free survival [5,6]. This is the case for patients with cancers that have prescribed precision medicine guidelines or when patients are screened for clinical trials with companion diagnostics, but this does not include every cancer patient currently.

In real-world clinical practice, histological biospecimens are commonly preserved as formalin-fixed paraffin-embedded (FFPE) samples. This is because FFPE tissue is essential for histological diagnosis and immunohistochemical staining. Furthermore, FFPE specimens provide advantages in terms of storage over fresh-frozen (FF) or cryopreserved (CP) samples, including the capability to be stored at room temperature for extended periods and convenient accessibility [7,8,9,10]. Thus, the main source of cancer biospecimen for NGS has been FFPE tissues. Nevertheless, the formalin-fixation and paraffin-embedding process leads to crosslinking, fragmentation, and degradation of DNA. These changes may adversely affect both the quantity and quality of DNA extracted from FFPE tissues, hence reducing the reliability of NGS when utilizing FFPE specimens [11,12]. Not only is DNA impacted by FFPE, but prior studies using RNA sequencing and real-time polymerase chain reaction (PCR) revealed that FFPE biospecimens are inferior to FF samples for those NGS analyses [13,14,15]. However, there have only been a few matched pair analyses of frozen tissue against FFPE, and in none of which, as far as we are aware, has the freezing process preserved the viability of the tumor tissue by slow freezing, also known as cryopreservation. The largest study to conduct a matched tissue comparison of flash-frozen tissue versus FFPE was performed as part of the 100,000-genome project that included 52 patient samples. Their study was limited in the accrual of patients because the research team failed to acquire frozen samples for 48% of the patients, underlining the logistical challenges of obtaining properly frozen tissue. Furthermore, equally concerning was that 16% of patient samples failed due to the poor quality of the FFPE DNAs during extraction and library preparation. The results for the remaining matched samples demonstrated 71% concordance in single nucleotide variants, 44% concordance in detecting copy number alternations, but 98% concordance in clinically actionable variants [10].

In this study, we aimed to compare the outputs of WGS by using paired FFPE and CP samples in terms of the sequencing quality control metrics and the number of identified variants. Similarly to other studies that have performed matched comparisons, we have demonstrated significant deficiencies in DNA from FFPE tissues that impact the quality control of the samples in WGS pipelines. However, we also demonstrate that the artifacts introduced by FFPE processing impact actionable variants, which can affect precision medicine decisions.

## 2. Results

### 2.1. Sample Types

Tumor samples were pulled from a commercially available biobank. The tumors represent 32 different histological types taken from fourteen unique primary locations. The details about the tumors and the patients from which they were taken are provided in Table 1. Each sample histological subtype was reviewed and confirmed by a board-certified pathologist. There was an overrepresentation of gynecological specimens, which skewed the percentage of female patients. Regardless, the sample procurement represents a diverse set of cancers and can be considered pan cancer.

The matched 50 tumor samples were processed between 2018 and 2020. DNA was extracted from matched sets in 2023 to conduct a comparative analysis between FFPE and CP tissue, whereupon whole-genome next-generation sequencing was performed to determine variant call concordance between the matched samples. It was hypothesized that CP tissue would demonstrate superior pre- and post-sequencing metrics.

### 2.2. Presequencing Metrics

The presequencing metrics measured for comparative analysis between FFPE and CP were DNA quantity and quality after extraction. Furthermore, the DNA concentration and DNA fragment size were measured after the library generation. It was observed that the average quantity of DNA obtained from CP tissue was seven times more than FFPE tissue: 85.2 ng/µL compared to 12.5 ng/µL, *p* value < 0.001 (Figure 1A). The DNA Integrity Number (DIN) was measured on an Agilent 2200 TapeStation (Agilent, Santa Clara, CA, USA) to ascertain the quality of the DNA fragments prior to library preparation. The CP tissue demonstrated a mean value of 8.4, while the FFPE exhibited a DIN of 4.7, *p* value < 0.001 (Figure 1B). All samples, regardless of the DIN score and quantity, were prepared for whole-genome sequencing by creating genomic libraries using TruSeq Nano Library Prep Kits (Illumina, San Diego, CA, USA). The concentration of DNA was measured again as well as the DNA library fragment size. In both metrics, the CP tissue again demonstrated superior results when compared to the FFPE tissue. Despite matched samples being normalized with 200 ng of DNA for the library prep kit, the CP tissue exhibited a DNA library concentration of 340.0 ng/µL, whereas the FFPE displayed a concentration of 137.8 ng/µL, *p* value < 0.001 (Figure 1C). The fragment size after the libraries were generated showed that CP tissue was statistically greater than FFPE tissue at 644.6 bp compared to 444.1 bp, *p* value < 0.001 (Figure 1D). To evaluate if the time spent in storage had an impact on the quantity of DNA extracted from the tissue, a correlation was graphed, and Pearson’s correlation test was performed. The Pearson correlation coefficient for the CP tissue was −0.41 (*p* value = 0.0032). This suggests a relationship that as CP tissue is stored beyond three years, the quantity of gDNA decreases. This was confirmed by a one-way ANOVA with a *p* value of 0.0044. The means for samples collected 3, 4, and 5 years ago were, respectively, 122.8 ng/µL, 56.6 ng/µL, and 60.0 ng/µL. gDNA from samples stored for 4 and 5 years was statistically indistinguishable (Figure 1E). Conversely, FFPE tissue had a low Pearson correlation coefficient of 0.14 (*p* value = 0.33), which was not statistically significant. Similarly, a one-way ANOVA did not show a difference between groups (*p* value = 0.44), and the means demonstrated no statistical difference: 3 years = 11.7 ng/µL, 4 years = 21.1 ng/µL, and 5 years = 21.1 ng/µL (Figure 1E). The lack of differences and overall poor quantity suggest that the quality of the tissue was compromised before the gDNA extraction took place as early as 3 years post-tissue collection.

### 2.3. Whole-Genome Sequencing Metrics

The matched DNA samples from CP tissue (n = 50) and FFPE tissue (n = 50) were processed for whole-genome sequencing on a next-generation NovaSeq sequencer. There were several quality metrics that were compared between CP and FFPE to determine the superiority of CP tissue over FFPE tissue. The mean read depth was measured between the two groups. The CP tissue’s DNA exhibited a mean of 54.2×, while the FFPE tissue’s DNA exhibited a mean of 34.6×, *p* value < 0.001 (Figure 2A). When comparing the percent of the genome that was represented with mapped reads that were >15×, the mean values were similar between CP and FFPE with 99.7% and 99.3%; however, the statistical analysis still showed a significant improvement in CP tissue with a *p* value < 0.001 (Figure 2B). Despite having 35% more average total reads in the CP tissue (1.2 × 10^9^ reads) compared to FFPE tissue (8.9 × 10^8^), the FFPE tissue exhibited twice as many unmapped reads when compared to the CP tissue, 6.2 × 10^6^ and 3.6 × 10^6^, respectively. The percent mapping rate was very similar to the percent mapped (Figure 2C). The median insert size was the most significant difference between CP tissue and FFPE tissue, *p* value = 9.7 × 10^−25^. The CP value was 365.7 bp and the FFPE value was 234.6. While a 131 bp difference does not seem significant for downstream analysis, later it was determined that chromosomal structural variants were detected more in CP tissue because of this expanded base pair coverage on the median insert size (Figure 2D). One of the problems with PCR amplification during DNA library preparation is that it can introduce bias and inaccurate variant calling in the bioinformatic analysis. A surrogate marker for this bias introduced by PCR amplification is to measure the duplication rate of sequenced reads. Both sample types exhibited an acceptable duplication rate of <10% with a mean duplication rate of 6.3 for CP and 9.2 for FFPE (Figure 2E). Statistically, there was no difference. Upon analysis of discordant read proportion, the mean for CP tissue was slightly, yet significantly, lower at 1.1%, while the FFPE tissue mean was 1.9% discordant mapped reads, *p* value < 0.001 (Figure 2F). Five of the six sequencing metrics demonstrated that CP tissue was statistically superior when compared to FFPE tissue, and even for the one measure that was not statistically different, the trend favored CP tissue.

### 2.4. Variant Call Analysis

To further illustrate the differences between CP tissue whole-genome sequencing and FFPE tissue whole-genome sequencing, the variant alleles present in the matched samples were compared. Using a standard variant allele frequency (VAF) threshold of 5%, it was determined that the median point mutation overlap between CP and FFPE matched tissue was 43.5%. Further analysis upon changing the VAF threshold to eliminate rare allele variants improved the concordance linearly until a 20% VAF was reached and then peaked at 30% VAF, with the concordance between matched samples being 73.4% (Figure 3A–C). To investigate the impact that time in storage had on overlapping variants between CP and FFPE tissue, a correlation analysis was conducted. Groups were separated by the year they were collected (i.e., 2018, 2019, or 2020), which corresponds to 5 years, 4 years, and 3 years in storage. The analysis of variance demonstrated no statistical difference between the three groups (*p* value = 0.35), with corresponding means of concordance for the different time points being 46% for 3 years (n = 17), 42.5% for 4 years (n = 23), and 41.5% for 5 years (n = 10) (Figure 3D).

### 2.5. Tumor Mutational Burden Analysis

The previous analyses included all 50 samples with a matched comparison between CP and FFPE tissue. These analyses did not differentiate between germline and somatic variants found in the tissue. However, to perform an analysis of tumor mutational burden, germline variants needed to be identified and subtracted from the variant calls. Out of the 50 tumor samples analyzed for whole-genome sequencing, eight had accompanying stored blood for a germline analysis. Whole-genome sequencing was performed on those eight blood samples, and germline variants were subtracted from both CP and FFPE identified variants. The tumor mutational burden was statistically different between CP and FFPE tumors. CP tissue exhibited a median TMB of 6.4/Mb and FFPE tissue exhibited a median TMB of 13.7/Mb (Table 2). The total number of point mutations (PMs), including single nucleotide variants (SNVs), multinucleotide variants (MNVs), and small insertions and deletions (Indels), was almost double in the FFPE tissue when compared to the median value for CP tissue. The driver of these differences was not found primarily in the SNVs but in the MNVs and Indels.

The analysis of TMB at the chromosomal level revealed no statistical differences between CP and FFPE tissue. The structural variants of large deletions and duplications, inversions, and translocations were statistically indistinguishable; however, the trend showed greater detection in CP tissue, which is likely due to larger insert sizes and library DNA fragment sizes.

### 2.6. Cancer Gene Set

The present study focused on whole-genome sequencing in matched tumor samples processed by FFPE or cryopreservation. Previous studies have shown similar discordance between FFPE and frozen tissue across the entire genome but high concordance across cancer genes that impact diagnostic and therapeutic decisions. In this study, a cancer gene set was analyzed for concordance. By performing a Wilcoxon signed-rank test, we were able to compare the matched samples for concordance across a select cancer gene set. A summary of the comparison is provided in Table 3. The cancer gene variants are presented as medians with interquartile ranges; while inappropriate to perform statistics on the means, it was calculated that the FFPE tissue exhibited an average of 6.1 driver mutations per tumor, and for the CP tissue, it was 7.5 driver mutations per tumor. The median number of cancer gene variants is five in the FFPE tissue and six in the CP tissue, *p* value = 0.02. Upon investigation of overlapping variants between FFPE and CP tissue, the median number of concordant variants was four. Among the unique variants, the SNVs and SVs were statistically different, and the subtotal of unique variants was statistically different at 1 and 1.5 for FFPE and CP, respectively.

There were 194 counts of discordance between FFPE and CP tissue among the cancer gene list. The most repeated genes on this list were *TP53*, *PTPRB/D*, *SMARCA4*, *NOTCH1/2*, *KMT2C/D*, and *ARID1A/B.* Other notable genes with targeted therapies that exhibited discordance at least once among the cohort were *EGFR*, *ERBB2*, *VEGFA*, and *VHL* (Appendix A).

The differences between FFPE and CP tissue extend beyond the preanalytical variables and quality metrics after sequencing. The variant allele calls were most disparate at a VAF of 5%, and then the TMB was measured in a subset of samples with available germline variant data. The cancer gene analysis resulted in differences between FFPE and CP tissue in notable cancer genes with targeted precision medicine therapies. It is recommended that CP tissue be used for WGS to ensure accuracy in the efforts to pursue better patient outcomes with precision medicines.

Interestingly, a previous matched study comparing fresh-frozen colorectal cancer tissue to FFPE processed tissue suggested a high concordance between actionable cancer genes, ranging from 74 to 100% [11]. However, upon deeper evaluation of the formula to calculate concordance, the inclusion of true negatives skewed the concordance to be high. True negatives are not weighted equally to positives and do not occur at the same level of prevalence. We conducted a side-by-side comparison of the previous data and our data to demonstrate more comparable concordance rates between CP/FF and FFPE tissue. Instead of an average 94% concordance when including true negatives [11], the recalculated concordance was 56% (Appendix A), and compared to our study and focusing on the subset of cancer genes, we observed a 38% concordance (Appendix A). Equally important to concordance are values of sensitivity (true positives/true positives + false negatives) and positive predictive value (true positives/true positives + false positives). The mean sensitivity for FFPE in this study was 0.58 compared to 0.63 in the study by Gao et al. The mean positive predictive value (PPV) for FFPE in this study was 0.52 compared to 0.78 in the previously published study [11]. This underlines the importance of evaluating data without any skewing or bias so that the true clinical impact of FFPE can be determined.

## 3. Discussion

This study provides a comprehensive analysis of WGS outcomes from matched tumor samples preserved as either FFPE or CP specimens. This study is unique from other comparison studies because we are the first to do so with a CP tissue format instead of flash freezing. Across all evaluated parameters, CP tissue consistently demonstrated superior sequencing quality metrics compared to FFPE tissue. These findings hold significant implications in the era of precision oncology, where the fidelity of genomic data directly informs diagnostic accuracy, therapeutic decisions, and patient outcomes. Additionally, CP tissue opens up tissue testing capabilities because of the maintenance of cell viability.

One important assumption that was made in this study is that variants found in CP tissue are considered true positives. This assumption is based on studies comparing fresh tissue, FF, and FFPE, which demonstrated that while there were subtle differences between fresh and FF, they were statistically indistinguishable from each other, yet distinct from FFPE tissue [16]. When calculating concordance, sensitivity, and positive predictive values in Appendix A, we followed others in the field who reference FF and CP as true positives because of their molecular semblance of fresh tissue [11]. Concordance alone is an insufficient analysis of the impact that FFPE has on accurately identifying true cancer gene variants.

Evidence from large-scale population studies further contextualizes these results. Analyses from the 100,000 Genomes Project showed that, although FFPE samples exhibit inferior raw sequencing quality compared to fresh-frozen tissue, clinically actionable variants can still be recovered when FFPE-specific artifact signatures are computationally identified and filtered. To address this, the researchers developed an FFPE Impact Score—a metric designed to quantify the extent of fixation-related error signatures, thereby approximating the underlying biological signal [9]. However, this corrective framework is unnecessary when cryopreserved tissue serves as the sequencing substrate, since such samples preserve native DNA integrity without fixation artifacts. While the FFPE Impact Scoring can approach—but not fully achieve—the genomic “ground truth”, its performance remains constrained by read depth and modeling assumptions. Notably, this study did not examine variants with allele frequencies below 5%, an omission that is increasingly significant as accumulating evidence shows that a meaningful fraction of pathogenic alterations occur below this threshold [17]. Further research directly comparing FFPE and cryopreserved tissue in this low-frequency range is warranted and is becoming critically important for accurate genomic interpretation.

Despite these developments, the intrinsic limitations of FFPE remain clear. Formalin fixation introduces DNA crosslinking, fragmentation, and degradation, often leading to poor DNA quality control performance, increased sequencing artifacts, and reduced reliability in downstream analyses [7,8,9,18,19]. Our results confirm these prior observations [20,21,22] and build on previous comparative studies [10,11,23,24] by presenting one of the largest matched-sample analyses to date, reinforcing concerns regarding the suitability of FFPE for genomic applications.

Among quality metrics, the DNA Integrity Number (DIN) proved informative. CP samples exhibited significantly higher DIN values than FFPE, reflecting better preservation of intact genomic DNA. While alternative measures such as the DNA Quality Number (DQN) have been proposed, DIN remains a widely adopted and reproducible standard [25]. It was noted that some metrics demonstrated conflicting patterns of change in that the FFPE sample was better than the CP sample. Upon a deeper investigation, it was determined that the reversal of the trend that CP was superior to FFPE did not come from a consistently bad CP sample, which suggests that it was due to random errors introduced in the multistep handling of a tissue from resection to DNA extraction to sequencing. Each step has some inherent variability that could contribute to these rare inverse patterns. To overcome these random errors, we relied on a large sample size of 50 matched samples, and the consensus pattern of the means was that CP tissue was better across all metrics. Another possible explanation for the inverse patterns where FFPE was better than CP tissue is that tumors are heterogenous, and despite the pathological analysis of viable tissue performed, the majority of the tumor used for DNA extraction could have come from less viable or less cellular tissue. Furthermore, we investigated if the time in storage had any impact on the quality metrics. While DNA quantity appeared to decrease after 3 years in storage in CP tissue, there was no statistical difference in concordance across those tumors taken 3–5 years ago.

Importantly, the advantages of CP extended beyond quality control metrics to sequencing outcomes themselves. CP-derived DNA yielded higher concentrations, longer fragment sizes after library preparation, superior read mapping rates, and greater concordance in variant allele detection. These properties are particularly beneficial for detecting structural variants and oncogenic driver mutations, which require high-quality, long-fragment DNA. These observations are consistent with our prior work and with other reports showing comparable trends [12].

While CP offers clear technical advantages, FFPE continues to dominate routine clinical practice because of its logistical convenience and long-term stability at room temperature [26,27]. However, newer preservation strategies, such as HypoThermosol and RNAlater, enable short-term refrigerated storage prior to cryopreservation and represent feasible alternatives when immediate flash freezing is not possible [28,29]. Adoption of these methods could expand access to CP-quality biospecimens without disrupting existing workflows. The current workflow of providing FFPE tissue to anatomical pathologists is still an essential step of the clinical diagnostic workup as FFPE tissue provides superior samples for histopathology. However, considerations to process as little as possible for FFPE evaluations should be prioritized, which will allow for multiomic molecular analysis of the tumor tissue. Manageable workflows that emphasize viable tissue preservation will make more translational research and clinical assays in personalized medicine available. In He et al.’s study, the authors detailed steps that maximize cell viability and make the generation of patient avatars for drug testing possible [30]. Tissue management practices that maximize cell viability and molecular quality rely on coordination between surgeons, pathologists, and clinical researchers that minimize warm ischemia time and follow NCCN guidelines for fixation times [31].

From a clinical perspective, the artifacts introduced by FFPE carry significant implications. We observed increased frequencies of MNVs and small Indels in FFPE-derived data, likely attributable to chemical modifications and DNA fragmentation caused by fixation. Such artifacts can generate both false-positive and false-negative variant calls, undermining the reliability of clinical sequencing [9,10,11]. Currently, the field of variant analysis is limited to variants with a high variant allele frequency. As seen in Figure 3, the concordance increased when the variant threshold was limited to only those >20%. The higher concordance at higher VAFs is likely due to random artifacts that occur at a frequency of less than 5%. The prevalence of artifacts occurring <5% obscure the true variants that occur at low allele frequencies. Conversely, CP samples enabled higher detection rates of clinically relevant mutations, suggesting that FFPE preservation may obscure critical genomic alterations in cancer genes.

Moreover, the preservation-dependent biases we identified in DNA analyses are echoed in transcriptomic studies, where FFPE-derived RNA displays greater degradation, higher proportions of intronic and intergenic reads, and more discordant variant calls compared with fresh or frozen material [16]. Collectively, these findings underscore the cross-platform consequences of fixation and the broader value of cryopreservation for multiomic investigations.

Taken together, our results emphasize that the tissue preservation method is a critical determinant of sequencing reliability. While FFPE will remain indispensable for histopathology, cryopreservation offers substantial advantages for genomic applications and warrants broader integration into both research and clinical workflows.

## 4. Materials and Methods

### 4.1. Study Design

Samples were provided by Specicare (Specicare, Gainesville, GA, USA), a biospecimen management company with an emphasis on cryopreservation tumor tissue to ensure maximum cell viability and unaltered molecular signatures in the cancer. Specicare collected tumor tissue from consented patients presenting for surgical removal of tumor tissue during standard of care for malignancy. The samples were split into either formalin-fixed tumor tissue or fresh tumor tissue stored in HypoThermosol (Stem Cell Technologies, Vancouver, BC, Canada) solution, transported at 4 °C, and, within 24 h, underwent a media exchange before undergoing our slow freezing protocol. FFPE samples were fixed for a minimum of 18 h before dehydrating and embedding in wax blocks. FFPE samples were stained with hematoxylin and eosin and evaluated by a certified board pathologist for accurate diagnosis, percent tumor (>20%), and viability, and the samples with severe necrosis (>25%) were considered unacceptable for use in the study. All patient samples were deidentified with unique identification numbers provided by Specicare. Fifty deidentified human patient samples were processed in parallel to be formalin-fixed, embedded in wax or cryopreserved, and stored in liquid nitrogen. Samples were collected between 2018 and 2020. Cryopreserved tissues were stored in liquid nitrogen, and FFPE blocks were stored at room temperature in a dark, climate-controlled room until shipment to Inocras for further analysis. WGS of each CP and FFPE sample was conducted by CancerVision (Inocras, San Diego, CA, USA), a proprietary comprehensive genomic profiling product [32].

### 4.2. WGS Methods

#### 4.2.1. DNA Extraction and Sequencing

Tumor DNA extraction from both CP and FFPE specimens was performed by Inocras (San Diego, CA, USA) using MagMax FFPE DNA/RNA Ultra Kit for FFPE (Themo Fisher Scientific, Waltham, MA, USA). For CP samples, 50 mg of tissue was provided, and for the FFPE samples, two 50 µm scrolls taken from the thickest part of the tumor were provided. The FFPE material was transferred to a AutoLys tube (Thermo Fisher Scientific, Waltham, MA, USA). A buffer was added to the tube and then incubated in a heat block until the paraffin was melted. The tube was then centrifuged to remove the paraffin debris. After manual deparaffinization, the solution was transferred to the KingFisher Apex (Thermo Fisher Scientific, Waltham, MA, USA) for automated nucleic acid extraction. An RNase digestion step was included in the DNA extraction. DNA from FFPE or CP tissue was assessed with both Qubit (Thermo Fisher Scientific, Waltham, MA, USA) and TapeStation (Agilent, Santa Clara, CA, USA) to perform an assessment of the amount of degradation in the sample by reviewing the size of the DNA peak and evaluating the DIN value. These results were used to determine the acceptability of the sample. If the value was below 2 ng/µL for the concentration and below 3.5 for a DIN value, the sample was not acceptable for library prep and sequencing. Both FFPE samples and CP tissue samples were fragmented enzymatically as part of library preparation. The CancerVision system was utilized to perform the analysis and interpretation of WGS. DNA libraries were generated using Watchmaker Genomics DNA Library Prep Kit (Watchmaker Genomics, Boulder, CO, USA) with 200 ng of DNA extracted from each sample used as the input. DNA concentration and DIN measurements were obtained using a TapeStation system (Agilent Technologies, Santa Clara, CA, USA). The libraries were then sequenced on the Illumina NovaSeq 6000 or NovaSeq X Plus platform (Illumina Inc., San Diego, CA, USA) with an average depth of 40× for tumor samples in WGS.

#### 4.2.2. Bioinformatic Analysis of WGS Data

The obtained sequences were aligned to the human reference genome (GRCh38) using the BWA-MEM v1.0.4 algorithm [33]. SAMBLASTER v0.1.26 was utilized to remove PCR duplicates [34]. Using Strelka2 v2.9.10 and Mutect2–GATK v4.2.0.0, small variant calling was performed, and the variants were combined. Manta v1.6.0 was utilized to detect structural variations [35]. Identified variants were annotated using VEP release 107 [36] and manually inspected and curated using Inocras’ proprietary genome browser. Sequenza-based analysis yielded estimates of tumor characteristics, including tumor cell fraction (TCF), tumor cell ploidy, and segmented copy number alteration profiles [37].

The process of calling point mutations employed stratified cutoffs dependent on the type of mutation (hotspot vs. non-hotspot) [38]. For hotspot mutations, a minimum variant allele frequency (VAF) of 1% and at least 2 variant allele reads in the tumor samples were required. Non-hotspot mutations were subject to more stringent criteria: a VAF of at least 5% and a minimum of 3 variant allele reads. These cutoffs provide a balance between sensitivity, which ensures the capture of true mutations, and specificity, which minimizes false-positive results. The detection of copy number variations (CNVs) prioritized smaller copy number changes over entire chromosomal arm changes in copy number because smaller copy number changes are usually more biologically meaningful and easier to use to confirm true signals. This guarantees a high level of confidence in identifying true amplification events. Biallelic deletions were identified when the copy numbers of both major and minor alleles at the same loci were equal to zero.

Tumor mutational burden (TMB) was calculated as the sum of single nucleotide variants (SNVs) and Indels divided by an effective genome size (i.e., 2,862,010,428 bp) and then normalized to SNVs per megabase (Mb) [39]. Microsatellite instability (MSI) was determined using a modified algorithm based on MSIsensor [24], and the HRD score was estimated using a refined version of HRDetect, optimized for improved performance in FFPE tissue [40].

### 4.3. Statistical Analysis

The Shapiro–Wilk test was used to evaluate if the variables are normally distributed. Nonparametric continuous variables are presented as medians with an interquartile range (IQR), unless otherwise noted. Wilcoxon’s signed-rank test was used to compare the nonparametric continuous variables between CP and FFPE samples. Two-sided *p* values were calculated for all analyses. A *p* value less than 0.05 was considered statistically significant. All data were analyzed using Python version 3.7 (Python Software Foundation, Wilmington, DE, USA) and R statistics version 4.2.0 (R Foundation, Vienna, Austria).

### 4.4. Bias Mitigation

Tumor tissue samples were provided by Specicare Inc., a commercial biobank, under informed patient consent. However, once deidentified, all analytical work, including DNA extraction, library preparation, sequencing, and bioinformatics processing, was performed by Inocras Inc. The design, execution, and interpretation of data were led by academic investigators at Brigham Young University. While employees of Specicare were involved in sample collection and manuscript drafting, they did not participate in the laboratory work or primary data analysis. To ensure objectivity, the analysis included predefined and standardized pipelines, with blinded sample identifiers during sequencing and variant calling. Moreover, statistical analyses were conducted independently by the academic team to reduce potential confirmation bias.

## 5. Conclusions

This study underscores the central role played by biospecimen preservation in enabling accurate and reproducible genomic analyses. Although FFPE remains the gold standard for traditional pathology, its limitations in sequencing applications are increasingly evident. Cryopreservation, in contrast, consistently preserves DNA integrity and yields superior WGS performance.

Given these findings, further matched-sample studies are warranted to re-evaluate tissue preservation standards for molecular oncology. In the interim, prioritizing CP specimens where feasible may enhance the reliability of genomic analyses, particularly for applications requiring the detection of structural variants and clinically actionable mutations. The integration of CP biospecimens into routine workflows has the potential to improve genomic profiling, refine therapeutic decision-making, and ultimately advance patient outcomes in precision medicine.

## Figures and Tables

**Figure 1 ijms-26-11038-f001:**
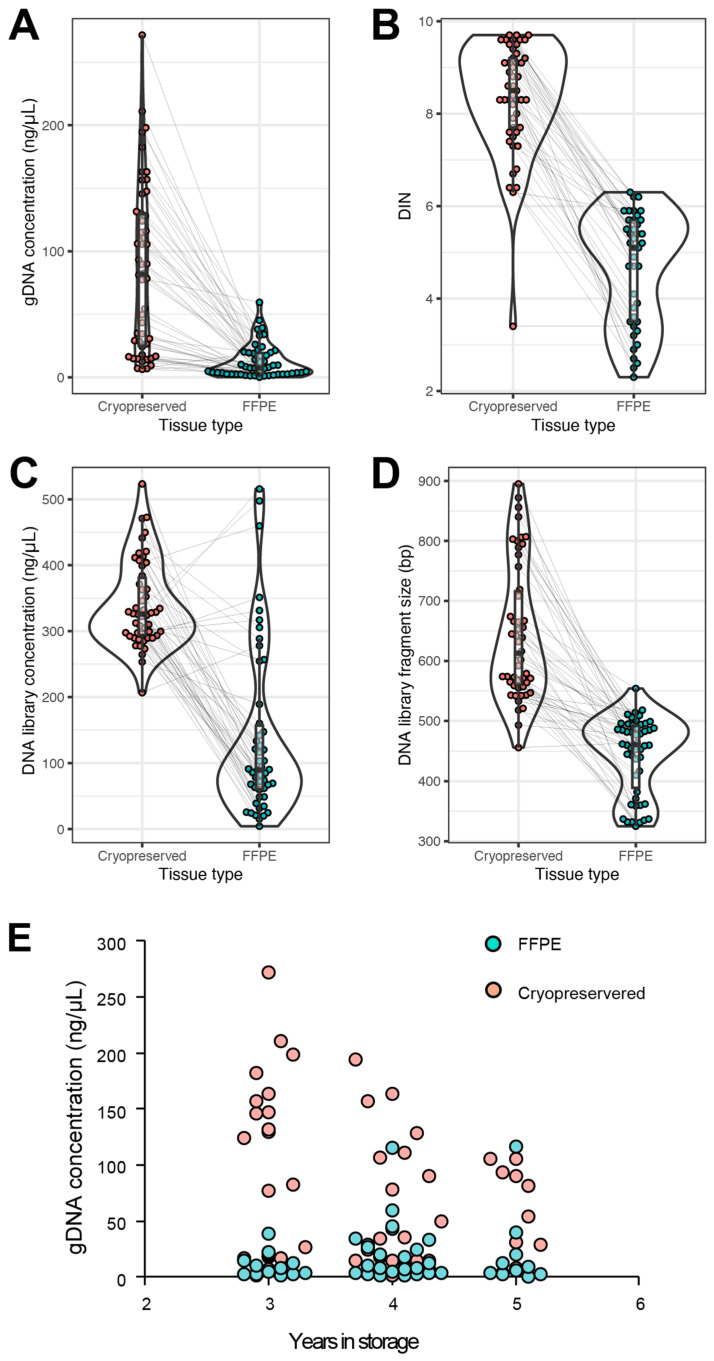
**Presequencing quality metrics.** Cryopreserved tissue (red) was compared to FFPE (green) in four different performance metrics: genomic DNA concentration (**A**), DIN value (**B**), DNA library concentration (**C**), and DNA library fragment size (**D**). In each comparison, the statistical analysis demonstrated that CP tissue was superior to FFPE tissue; all *p* values < 0.001. gDNA concentration correlation to time spent in storage (**E**). Gray lines connect matched samples between cryopreserved and FFPE.

**Figure 2 ijms-26-11038-f002:**
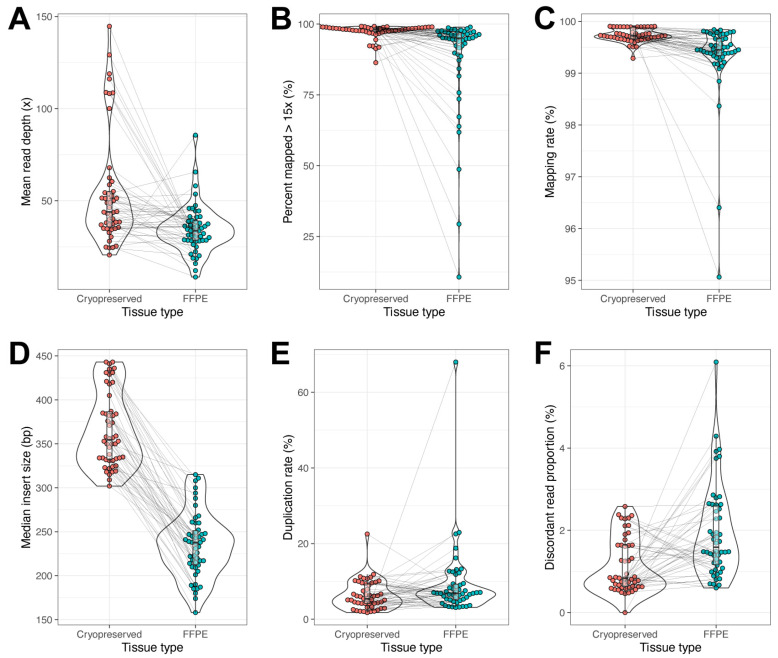
**Whole-genome sequencing quality metrics.** Cryopreserved tissue (red) was compared to FFPE (green) in six different performance metrics: mean read depth (**A**), percent mapped reads > 15× (**B**), percent mapping rate (**C**), mean insert size (**D**), percent duplication rate (**E**), and percent discordant read proportion (**F**). In each comparison, the statistical analysis demonstrated that CP tissue was superior to FFPE tissue; all *p* values < 0.001. Gray lines connect matched samples between cryopreserved and FFPE.

**Figure 3 ijms-26-11038-f003:**
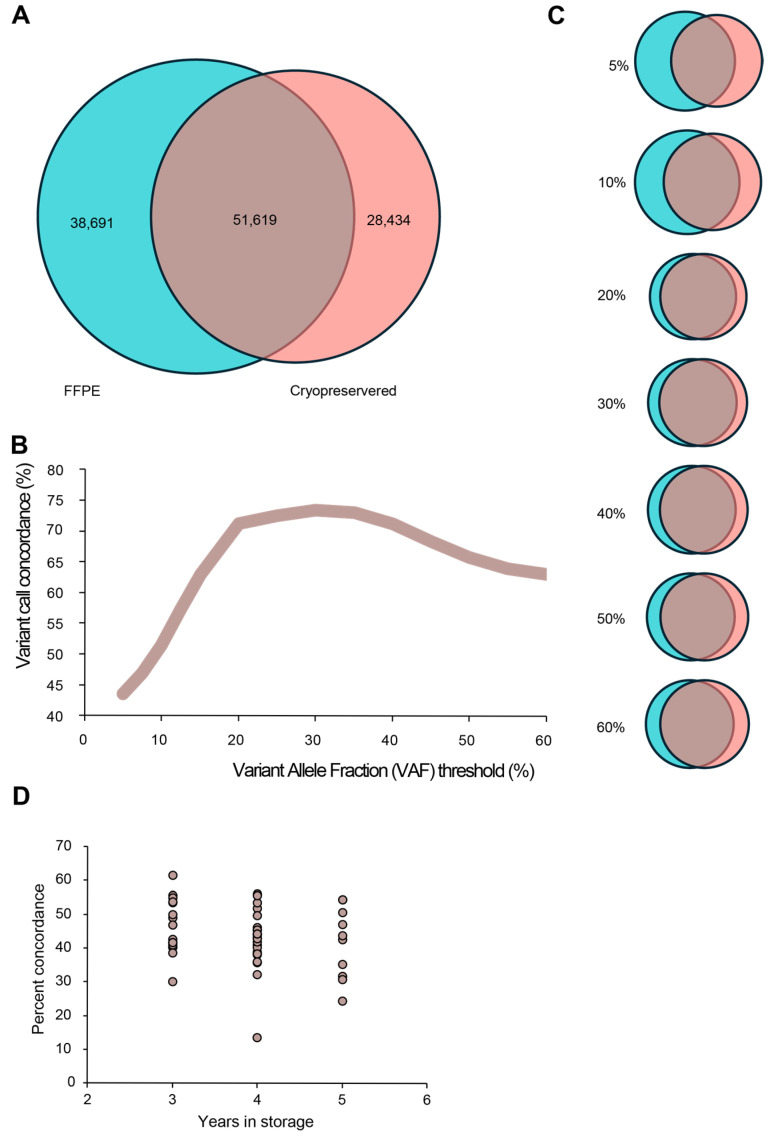
**Variant allele concordance between CP and FFPE tissue.** Cryopreserved tissue (red) was compared to FFPE (green) in terms of variant allele calls, and concordance was measured at the standard VAF threshold of 5%. The median number of concordant variant alleles was 51,619 calls (brown), and this equated to 43.5% concordance (**A**). Concordance was measured at increasing thresholds of VAF (5–60%). The correlation and graphical representation of this relationship between VAF and concordance is provided in (**B**,**C**). The correlation between time in storage and percent concordance between CP and FFPE tissue is represented in (**D**) (n = 50).

**Table 1 ijms-26-11038-t001:** Patients and tumor characteristics.

**Patient (n = 50)**			
Age, year			63.6 ± 11.8
Sex assigned at birth	Male Female		18 (36.0%) 32 (64.0%)
**Tumor (n = 50)**			
Specimen Source	n	(%)	Histological type
Ovary	14	28%	Serous carcinoma 7 Clear cell carcinoma 2 Papillary serous adenocarcinoma 1 Carcinoma 1 Carcinosarcoma 1 Benign stromal tumor 1 Undetermined 1
Uterus	7	14%	Carcinosarcoma 3 Adenocarcinoma 1 Endometrioid carcinoma 1 Leiomyosarcoma 1 * Undetermined 1
Kidney	6	12%	Renal clear cell carcinoma 4 Oncocytoma 1 Urothelial cancer 1 *
Liver	5	10%	Hepatocellular carcinoma 2 Adenocarcinoma 2 * Poorly differentiated carcinoma 1
Colon	5	10%	Adenocarcinoma 3 Granulosa cell tumor 1 * Papillary serous carcinoma 1 *
Adrenal gland	2	4%	Cortical carcinoma 1 Diffuse large B-cell lymphoma 1 *
Lung	2	4%	Epithelioid mesothelioma 1 Leiomyosarcoma 1 *
Peritoneum	2	4%	Ovarian adenocarcinoma 1 * Ovarian serous carcinoma 1 *
Breast	2	4%	Invasive ductal carcinoma 1 Ovarian adenocarcinoma 1 *
Bone	1	2%	Prostate adenocarcinoma 1 *
Thyroid	1	2%	Papillary carcinoma 1
Pelvis	1	2%	Liposarcoma 1
Stomach	1	2%	Gastrointestinal stromal tumor 1
Appendix	1	2%	Signet ring carcinoma 1

* Metastatic disease.

**Table 2 ijms-26-11038-t002:** Tumor mutational burden.

	FFPE	CP	*p* Value *
TMB, mut/Mb	13.7 (12.2–16.0)	6.4 (5.5–8.5)	0.02
PM (SNV + MNV + IND)	39,613 (34,839–45,786)	20,339 (16,638–24,959)	0.02
SNV			
Total	21,998 (16,789–26,046)	10,660 (8373–15,675)	0.10
C>A	2699 (2169–3234)	4492 (2773–4692)	0.31
C>G	1602 (1166–2068)	568 (262–1189)	0.11
C>T	5914 (4563–8282)	2084 (1328–3429)	0.14
T>A	2852 (1501–3310)	552 (449–893)	0.054
T>C	3769 (2459–4967)	1275 (950–1721)	0.04
T>G	2937 (2632–3733)	1504 (1271–2734)	0.20
MNV	1076 (492–1660)	83 (67–134)	0.008
Indel	18,174 (13,704–20,919)	8871 (8199–11,231)	0.02
SV			
Total	21 (13–62)	38 (12–81)	0.94
Deletion	4 (2–8)	12 (2–24)	0.11
Duplication	2 (0–3)	4 (3–14)	0.08
Inversion	11 (2–49)	10 (2–15)	0.15
Translocation	6 (4–7)	8 (5–27)	0.46

FFPE, formalin-fixed paraffin-embedded; CP, cryopreserved; SNV, single nucleotide variant; Indel/IND, insertion–deletion; SV, structural variant. Note: Data are presented as median (interquartile range). * by Wilcoxon signed-rank test.

**Table 3 ijms-26-11038-t003:** Variant number of cancer genes.

Variant Types	FFPE	CP	*p* Value *
Overall			
SNV	3 (2–4)	3 (2–5)	0.02
Indel	0 (0–1)	0 (0–1)	0.82
SV	1 (1–2)	1 (1–3)	0.007
CNV	0 (0–1.25)	0 (0–1)	0.58
Subtotal	5 (4–8)	6 (4–10.25)	0.02
Overlapped			
SNV	2.5 (1–4)		
Indel	0 (0–1)		
SV	1 (0–2)		
CNV	0 (0–0)		
Subtotal	4 (3–6)		
Unique			
SNV	0 (0–0)	0 (0–0)	0.02
Indel	0 (0–0)	0 (0–0)	1.00
SV	0 (0–0)	0 (0–0)	0.001
CNV	0 (0–0)	0 (0–0)	0.82
Subtotal	1 (1–2)	1.5 (0–2)	0.009

FFPE, formalin-fixed paraffin-embedded; SNV, single nucleotide variant; Indel, insertion–deletion; SV, structural variant; CNV, copy number variant. Note: Data are presented as median (interquartile range). * by Wilcoxon signed-rank test.

## Data Availability

The original contributions presented in this study are included in the article/Appendix A. Further inquiries can be directed to the corresponding authors.

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
