# Peer review of "Cryopreserved Tissue Biospecimens Offer Superior Quality for Whole-Genome Sequencing of Various Cancers Compared to Paired Formalin-Fixed Paraffin-Embedded Tissues"

_ijms, 2025, doi:10.3390/ijms262211038_

Round 1

Reviewer 1 Report

Comments and Suggestions for Authors

Dixon and colleagues provides a comprehensive comparison between cryopreserved (CP) and formalin-fixed paraffin-embedded (FFPE) tumor biospecimens for whole genome sequencing (WGS) applications across multiple cancer types. The authors demonstrate significant advantages of CP samples in DNA integrity, sequencing quality, and variant detection accuracy. The study highlights that FFPE-induced artifacts can inflate tumor mutational burden and compromise the detection of clinically actionable mutations, reinforcing the need to re-evaluate biospecimen preservation strategies in precision oncology. Overall, the manuscript is well-written and clearly structured. The experimental design is sound, the statistical analyses are appropriate, and the results are convincingly presented. However, some methodological clarifications, deeper contextualization in the discussion, and minor presentation improvements would strengthen the paper’s scientific rigor and readability, as shown below:

A) While the study offers robust data supporting CP tissue superiority, the novelty relative to recent large-scale genomic studies (e.g., Basyuni et al., Nat Commun 2024; Robbe et al., Genet Med 2018) should be more explicitly defined. Clarify how this work advances beyond confirming previously known FFPE limitations.

B) The reported 43.5% median variant overlap (rising to 73.4% at 30% VAF) is striking. The authors should provide a clearer explanation of how FFPE artifacts versus true biological variation contribute to this discordance, perhaps with visual examples (e.g., specific variant profiles).

C) While technical superiority is well demonstrated, the discussion could more strongly connect these findings to practical implications in clinical sequencing pipelines, particularly how cryopreservation might be integrated into current diagnostic workflows despite its logistical challenges.

D) Given the involvement of Specicare Inc. and Inocras Inc. in data generation, the manuscript would benefit from clearer delineation of data ownership, analysis independence, and measures to mitigate potential bias.

E) Include a graphical summary of key findings (e.g., workflow schematic comparing FFPE vs CP pipelines).

Author Response

Comment 1: 

Dixon and colleagues provides a comprehensive comparison between cryopreserved (CP) and formalin-fixed paraffin-embedded (FFPE) tumor biospecimens for whole genome sequencing (WGS) applications across multiple cancer types. The authors demonstrate significant advantages of CP samples in DNA integrity, sequencing quality, and variant detection accuracy. The study highlights that FFPE-induced artifacts can inflate tumor mutational burden and compromise the detection of clinically actionable mutations, reinforcing the need to re-evaluate biospecimen preservation strategies in precision oncology. Overall, the manuscript is well-written and clearly structured. The experimental design is sound, the statistical analyses are appropriate, and the results are convincingly presented. However, some methodological clarifications, deeper contextualization in the discussion, and minor presentation improvements would strengthen the paper’s scientific rigor and readability, as shown below:

Response 1:

We are grateful to our reviewer who was able to assess the importance of our findings and add critical insights that have strengthened our manuscript and the manner in which we can portray this data to the readership of IJMS. 

Comment 2: 

A) While the study offers robust data supporting CP tissue superiority, the novelty relative to recent large-scale genomic studies (e.g., Basyuni et al.Nat Commun2024; Robbe et al.Genet Med2018) should be more explicitly defined. Clarify how this work advances beyond confirming previously known FFPE limitations.

Response 2: 

The first paragraph in our discussion has updated statements that now reads:

“This study provides a comprehensive analysis of WGS outcomes from matched tumor samples preserved as either FFPE or CP specimens. This study is unique from other comparison studies because we are the first to do so with a CP tissue format instead of flash freezing. Across all evaluated parameters, CP tissue consistently demonstrated superior sequencing quality metrics compared to FFPE tissue. These findings hold significant implications in the era of precision oncology, where the fidelity of genomic data directly informs diagnostic accuracy, therapeutic decisions, and patient outcomes. Additionally, CP tissue opens tissue testing capabilities because of the maintenance of cell viability.”

Comment 3: 

B) The reported 43.5% median variant overlap (rising to 73.4% at 30% VAF) is striking. The authors should provide a clearer explanation of how FFPE artifacts versus true biological variation contribute to this discordance, perhaps with visual examples (e.g., specific variant profiles).

Response 3: 

We have added to our discussion section comments about the impact of artifactual variants and why concordance was strikingly higher above a VAF of 20%.

“Currently, the field of variant analysis is limited to variants with a high variant allele frequency. As seen in Figure 3, the concordance increased when the variant threshold was limited to only those > 20%. The higher concordance at higher VAFs is likely due to random artifacts that occur at a frequency of less than 5%. The prevalence of artifacts occurring < 5% obscure the true variants that occur at low allele frequencies.”

Comment 4:

C) While technical superiority is well demonstrated, the discussion could more strongly connect these findings to practical implications in clinical sequencing pipelines, particularly how cryopreservation might be integrated into current diagnostic workflows despite its logistical challenges.

Response 4:

We have added to our discussion section comments about workflows that will permit tissue management that maximizes information obtained from it.

“The current workflow of providing FFPE tissue to anatomical pathologists is still an essential step of the clinical diagnostic workup as FFPE tissue provides superior samples for histopathology. However, considerations to process as little as possible for FFPE evaluations should be prioritized, which will allow for multiomic molecular analysis of the tumor tissue. Manageable workflows that emphasize viable tissue preservation will make available more translational research and clinical assays in personalized medicine. In a study by He et al., the authors detailed steps that maximize cell viability and make possible the generation of patient avatars for drug testing [30]. Tissue management practices that maximize cell viability and molecular quality rely on coordination between surgeons, pathologists, and clinical researchers that minimize warm ischemia time and follow NCCN guidelines for fixation times [31].

Comment 5:

D) Given the involvement of Specicare Inc. and Inocras Inc. in data generation, the manuscript would benefit from clearer delineation of data ownership, analysis independence, and measures to mitigate potential bias.

Response 5:

We have added a new section in our material and methods to address these concerns.

4.4 Bias Mitigation

“Tumor tissue samples were provided by Specicare Inc., a commercial biobank, under informed patient consent. However, once deidentified, all analytical work, including DNA extraction, library preparation, sequencing, and bioinformatics processing, was performed by Inocras Inc. The design, execution, and interpretation of data were led by academic investigators at Brigham Young University. While employees of Specicare were involved in sample collection and manuscript drafting, they did not participate in the laboratory work or primary data analysis. To ensure objectivity, the analysis included predefined and standardized pipelines, with blinded sample identifiers during sequencing and variant calling. Moreover, statistical analyses were conducted independently by the academic team to reduce potential confirmation bias.”

Comment 6:

E) Include a graphical summary of key findings (e.g., workflow schematic comparing FFPE vs CP pipelines).

Response 6:

A graphical summary is now provided to overview our process.

Reviewer 2 Report

Comments and Suggestions for Authors

My expertise lies in tissue handling and preservation rather than genomic sequencing. My comments therefore will not focus on sequencing-related details.

In this study, the authors show that cryopreserved tissue biospecimens are better for genome sequencing than matched formalin-fixed samples in all evaluated parameter like genomic DNA concentration, DNA Integrity Number, DNA library concentration, and DNA library fragment size. These findings are of great interest since personalized treatments will become increasingly important in cancer medicine. The study is well conducted and the results are shown in a clear way.

One issue that I would like the authors to address: I think it is noteworthy that the cryopreserved sample with the lowest mean read depth corresponds to the formalin-fixed sample with the highest mean read depth. For gDNA concentration, it seems that the highest value in cryopreserved tissue corresponds to (one of) the lowest values in Formalin fixed tissue. One would assume, that the samples with the highest values from one preservation technique are from the same patient as the samples of (about) the highest values from the other preservation technique, just one order of magnitude lower/higher. However, in nearly all parameters it seems that a few samples show opposing ranks. Although most paired samples behave as expected, some show a striking difference. Maybe the authors can disclose whether it is the same sample in all occurences and provide an educated guess for the possible reason?

Author Response

Comment 1:

My expertise lies in tissue handling and preservation rather than genomic sequencing. My comments therefore will not focus on sequencing-related details.

In this study, the authors show that cryopreserved tissue biospecimens are better for genome sequencing than matched formalin-fixed samples in all evaluated parameter like genomic DNA concentration, DNA Integrity Number, DNA library concentration, and DNA library fragment size. These findings are of great interest since personalized treatments will become increasingly important in cancer medicine. The study is well conducted and the results are shown in a clear way.

One issue that I would like the authors to address: I think it is noteworthy that the cryopreserved sample with the lowest mean read depth corresponds to the formalin-fixed sample with the highest mean read depth. For gDNA concentration, it seems that the highest value in cryopreserved tissue corresponds to (one of) the lowest values in Formalin fixed tissue. One would assume, that the samples with the highest values from one preservation technique are from the same patient as the samples of (about) the highest values from the other preservation technique, just one order of magnitude lower/higher. However, in nearly all parameters it seems that a few samples show opposing ranks. Although most paired samples behave as expected, some show a striking difference. Maybe the authors can disclose whether it is the same sample in all occurences and provide an educated guess for the possible reason?

Response 1: 

We have commented on factors that could explain some inverse relationships between CP and FFPE samples. Our discussion section now includes the following statement:

“It was noted that some metrics demonstrated conflicting patterns of change in that the FFPE sample was better than the CP sample. Upon a deeper investigation, it was not a consistent sample, which suggests that it was due to random errors introduced in the multistep handling of a tissue from resection to DNA extraction, to sequencing. Each step has some inherent variability that could contribute to these rare inverse patterns. To overcome these random errors, we relied on a large sample size of 50 matched samples, and the consensus pattern of the means was that CP tissue was better across all metrics. Another possible explanation for the inverse patterns where FFPE was better than CP tissue is that tumors are heterogenous and despite the pathological analysis of viable tissue performed, the majority of the tumor used for DNA extraction could have come from less viable or less cellular tissue. Furthermore, we investigated if the time in storage had any impact on the quality metrics. While DNA appeared to decrease after 3 years in storage in CP tissue, there was no statistical difference in concordance across those tumors taken 3-5 years ago.”

Reviewer 3 Report

Comments and Suggestions for Authors

Manuscript: ijms-3943180

Recommendation: Major Revision

In this manuscript, the authors discuss the superiority of cryopreserved (CP) samples over formalin-fixed paraffin-embedded (FFPE) samples in whole-genome sequencing (WGS) analysis. They evaluated multiple parameters, including pre-sequencing DNA quality metrics, sequencing performance indicators, and variant-calling outcomes. The overall study design is reasonable, and the manuscript is logically written and easy for readers to follow. However, I have several concerns that should be addressed. The details are as follows:

  1. Regarding the WGS data, not all called variants necessarily represent true mutations. Is there a way for the authors to estimate or compare the false positive rates between the cryopreserved (CP) and formalin-fixed paraffin-embedded (FFPE) groups?
  2. In clinical practice, when physicians use WGS data to guide decision-making, tissue samples (either CP or FFPE) are often stored for less than a week or a month. Have the authors considered examining the effect of storage time on sequencing quality? For instance, do more recent FFPE samples (e.g., from 2020) perform better than older ones (e.g., from 2018)?
  3. As a valuable resource, FFPE tissue has its own advantages in clinical histopathology. Did the authors observe any aspects from the WGS data where FFPE samples perform better than CP samples? Moreover, have they considered that using both sample types in a complementary manner might help physicians make better clinical decisions?

Author Response

Comment 1:

In this manuscript, the authors discuss the superiority of cryopreserved (CP) samples over formalin-fixed paraffin-embedded (FFPE) samples in whole-genome sequencing (WGS) analysis. They evaluated multiple parameters, including pre-sequencing DNA quality metrics, sequencing performance indicators, and variant-calling outcomes. The overall study design is reasonable, and the manuscript is logically written and easy for readers to follow. However, I have several concerns that should be addressed. The details are as follows:

Regarding the WGS data, not all called variants necessarily represent true mutations. Is there a way for the authors to estimate or compare the false positive rates between the cryopreserved (CP) and formalin-fixed paraffin-embedded (FFPE) groups?

Response 1: 

We have added the following statement in our discussion section.

“One important assumption that was made in this study is that variants found in CP tissue are considered true positives. This assumption is based on studies comparing fresh tissue, FF, and FFPE, which demonstrated that while there were subtle differences between fresh and FF, they were statistically indistinguishable from each other, yet distinct from FFPE tissue [16]. When calculating concordance, sensitivity, and positive predictive values in Supplemental Tables 2 and 3, we followed others in the field who reference FF and CP as true positives because of its molecular semblance of fresh tissue [11]. Concordance alone is an insufficient analysis of the impact that FFPE has on accu-rately identifying true cancer gene variants.”

Comment 2: 

In clinical practice, when physicians use WGS data to guide decision-making, tissue samples (either CP or FFPE) are often stored for less than a week or a month. Have the authors considered examining the effect of storage time on sequencing quality? For instance, do more recent FFPE samples (e.g., from 2020) perform better than older ones (e.g., from 2018)?

Response 2:

We examined the impact of storage time on quantity of gDNA and have added a graph and analysis to figure 1.

“To evaluate if the time spent in storage had an impact on the quantity of DNA extracted from the tissue, a correlation was graphed, and a Pearson correlation test was performed. The Pearson correlation coefficient for the CP tissue was -0.41 (P value = 0.0032). This suggest a relationship that as CP tissue is stored beyond three years the quantity of gDNA decreases. This was confirmed by a one-way ANOVA with a P value of 0.0044. The mean for samples collected 3, 4, and 5 years ago were respectively, 122.8 ng/µL, 56.6 ng/µL, and 60.0 ng/µL. gDNA from samples stored for 4 and 5 years were statistically indistinguishable (Figure 1E). Conversely, FFPE tissue had a low Pearson correlation coefficient of 0.14 (P value = 0.33) that was not statistically significant. Similarly, a one-way ANOVA did not show a difference between groups (P value = 0.44) and the means demonstrated no statistical difference: 3 years = 11.7 ng/µL, 4 years = 21.1 ng/µL, and 21.1 ng/µL. The lack of differences and overall poor quantity suggests that the quality of tissue was com-promised before the gDNA extraction took place as early as 3 years post tissue collection.” 

We examined the impact of storage time on concordance and have added a graph and analysis to figure 3.

“To investigate the impact that time in storage had on overlapping variants between CP an FFPE tissue, a correlation analysis was conducted. Groups were separated by the year they were collected (i.e. 2018, 2019, or 2020), which corresponds to 5 years, 4 years, and 3 years in storage. The analysis of variance demonstrated no statistical difference between the three groups (P value 0.35) with corresponding means of concordance for the different time points being 46% for 3 years (n = 17), 42.5% for four years (n = 23), and 41.5% for five years (n = 10). “

Comment 3: 

As a valuable resource, FFPE tissue has its own advantages in clinical histopathology. Did the authors observe any aspects from the WGS data where FFPE samples perform better than CP samples? Moreover, have they considered that using both sample types in a complementary manner might help physicians make better clinical decisions?

Response 3:

“The current workflow of providing FFPE tissue to anatomical pathologists is still an essential step of the clinical diagnostic workup as FFPE tissue provides superior samples for histopathology. However, considerations to process as little as possible for FFPE evaluations should be prioritized, which will allow for multiomic molecular analysis of the tumor tissue.”

Round 2

Reviewer 1 Report

Comments and Suggestions for Authors

The authors provided a revised manuscript with significant improvements. All questions were answered satisfactorily.

Reviewer 3 Report

Comments and Suggestions for Authors

The authors have addressed all my concerns.